# Dialogic gathering of films. Promoting meaningful online interactions during COVID-19 confinement

**Maria Padrós-Cuxart**[1], **Roseli Rodrigues de Mello**[2], **Mimar Ramis-Salas**[3], **Elena Duque**[4]*

**1** Department of Teaching and Learning and Educational Organization, University of Barcelona, Barcelona, Spain, **2** Department of Pedagogical Theories and Practices, University of Sao Carlos, Sao Carlos, Brasil, **3** Department of Sociology, University of Barcelona, Barcelona, Spain, **4** Department of Theory and History of Education, University of Barcelona, Barcelona, Spain

* elenaduquesa@ub.edu

**Data Availability Statement:** Demographic data of participants is included within the manuscript. Data from closed-ended questions (Likert scale questions) are fully provided in the Supporting

## Abstract

A broad body of scientific literature exists on the effects that COVID-19 related confinement has had on the population: mental health problems, isolation, and problems concerning cohesion and employment, among others. However, there is a gap in the literature on the actions that reverse some of the effects generated during lockdown. This article collects the results of a study conducted with 53 people participating in a dialogic gathering of films (DGF) that was held online during two months of confinement. The data from the survey show that the development of this DGF generated improvements in 1) personal welfare and attitudes concerning the management of confinement, 2) living together and online relationships, 3) motivation and creativity in the professional domain, and 4) openness to a diversity of perspectives and realities, which improves the understanding, argumentation and positioning in social, scientific and ethical debates.

## Introduction

From the beginning of the COVID-19 pandemic, a large amount of research has been developed regarding the various consequences of lockdown. Socio-economic consequences [1–3], labour market repercussions [4–6], educational effects [7–10], influence on physical activity [11], eating habits and nutrition changes [12–15], domestic violence [16], elderly people care [17, 18] and environmental effects [2], among others, have been studied as a result of confinement. Prior to this situation, elements that can promote improved quality of life during confinement, such as physical activity [19, 20], had been studied. Mental health and psychological effects of the lockdown are among the most studied issues [21–29]. In this framework, social interaction is one of the elements that has been more affected by the confinement period, and it also plays a key role in benefitting people. Accordingly, the WHO [30] recommended staying connected and maintaining one's social networks to prevent isolation. To this end, the population used different strategies to help remain socially connected during confinement [31–33].

Information. Regarding the qualitative data, there are ethical restrictions on sharing it. The CREA ethical committee doesn't approve to make public all the qualitative data because the participants don't give their consent to put all their responses in an open repository. It is possible to deidentify the qualitative data and make it available upon request by contacting CREA ethical committee (crea@ub.edu).

**Funding:** Research Group on Education Overcoming Inequalities (GRESUD).

**Competing interests:** The authors have declared that no competing interests exist.

In Spain, a confinement period was initiated on 15 March 2020 due to the COVID-19 pandemic and continued until the so-called *new normality* started on 21 June 2020. During this period, a group of friends and colleagues, most of them university researchers, decided to organize a dialogic gathering of films (DGF henceforth) [34]. This paper presents the results of a study on the effects of the DGF, and it is developed in the framework of GRESUD at the University of Barcelona: a Research Group on Education Overcoming Inequalities with a specific research line focused on the study of dialogic gatherings. The research, some results of which are presented here, is framed in the project *Enlarge SEAs* (2019–2020) funded by the European Commission and focused on the extension in Europe of successful educational actions (SEAs) [35]. The dialogic gatherings are one of the SEAs that this project is promoting around European educational centres. The results of the research presented in this paper show the improvements that this specific DGF has had on its participants.

## Psycho-social effects of confinement and social interactions

The literature focusing on the psychological effects of confinement indicates that lockdown has generated negative effects such as anxiety and depression [36–38]. Various elements that can increase or decrease these negative psychological effects have also been identified. The perception of COVID-19 as a threat [39], the increase in exposure to social media [40] and the perceived impacts of the pandemic [21], among others, are associated with sadness, depression, anxiety and anger-hostility. Further research highlights that when COVID-19-related economic difficulties appear in low socio-economic environments, mental health problems can also be increased [41–43]. Other lines of research have highlighted that different individual profiles can be more vulnerable to suffering negative psychological effects. Research developed in Italy with 310 participants [44] highlights that women were more affected by psychological distress. Work developed in Spain with 1596 participants [45] also highlighted that women, together with students and the population with a lower level of economic income, suffer from more significant psychological impact and worse mental health. The research of Ogden [24] also includes elderly people as a vulnerable group. However, the work by Qian and Yahara [46] analysing how personality, morality and ideology influence people's mentality during COVID-19 shows that, regardless of individual differences, people with negative coping styles have higher levels of psychological distress than those with positive coping styles.

Social interaction has been identified in the literature as a preventive factor for negative psychological confinement effects. In a study developed in the UK, Ogden [24] identified that reduced satisfaction with current levels of social interactions was associated with a slowing perception of the passage of time. Another study in South Korea [47] analysed preventive health behaviours for psycho-social health. The results show that groups participating in culture and arts and social activities displayed characteristics highly associated with prevention. During the confinement period, people have also created new activities and strategies (or adapted those they usually had) to combat isolation [48–50]. According to Cohen and Cromwell [51], the COVID-19 problem can be addressed with creativity and innovation. Along this line, Bland [52] highlights that COVID-19 provides opportunities for collective co-creation and that in this particular period, people are more capable of being more creative. For example, various artistic manifestations [53–55] flourished during this period. Information and Communications Technology (ICT) has played a central role in keeping people connected [56], reducing stress and anxiety and/or alleviating depressed moods. Despite the difficulties of social connections, people have been attempting to overcome them. In fact, despite the lockdown and the necessary social distancing, new ways of social cohesion and solidarity have emerged in these crisis situations [57].

## Dialogic gatherings of films

Different sorts of dialogic gatherings have been defined, studied and analysed in previous research. Dialogic literary gatherings (DLGs) were first created in 1978 in an adult school in Spain [58, 59]. In recent years, DLGs have been extended to different contexts, and improvements in learning and social cohesion have been analysed extensively [60–64], including among vulnerable groups such as migrants, out-of-home child care, and people in correctional institutions [65–68]. Furthermore, new sorts of dialogic gatherings have been created, such as dialogic theological gatherings [69], dialogic pedagogical gatherings [70], dialogic musical gatherings, dialogic scientific gatherings [71], and dialogic feminist gatherings [72]. Furthermore, some educational centres that usually organize these dialogic gatherings have continued implementing them online during confinement with remarkable success [73, 74]. The common functioning that grounds all the dialogic gatherings is the collective construction of meaning and knowledge through an egalitarian dialogue [58] among all the participants.

This article focuses on the novel dialogic gatherings of films, which—despite having been created more than 20 years ago—have not been studied thus far. The DGF we study here does not represent all the DGFs held from their creation. Dialogic gatherings of films (DGFs) have evolved and innovated over time according to the circumstances in which they developed. This DGF in particular required specific creative elements for it to be developed in the particular context of confinement. Without losing the essence of dialogic gatherings in general, nor of the DGF in particular, the DGF we discuss in this paper was created specifically for current participation in an exceptional situation related to the confinement period lived in Spain due to COVID-19 from March to May 2020.

## Methods

This study was created following the communicative methodology [75–78] developed for more than 20 years through various European and national research projects. This methodology was chosen for this research because of its scientific recognition by the European Commission [79, 80] and because relevant research funded by the EC Framework Programmes has successfully used it in previous studies on dialogic gatherings [60, 81]. One of the main premises of this methodology is the egalitarian dialogue [58] that is held between researchers and participants, including participants in the entire research process [76, 82, 83], overcoming the interpretative hierarchy [84] and creating collective meaning [84, 85]. These characteristics become ideal for investigating DGFs because they allow researchers to establish a dialogue with participants throughout the research process and to join researchers' and participants' interpretations and analysis in a collective understanding of the functioning and benefits produced by DGFs. The communicative methodology targets social transformation, understood as having a positive impact in improving people's lives [75, 77]. This premise was essential in the selection of this methodology for the study of the DGF because the objective was just to study the benefits of DGFs and how they can contribute to improving people's lives during confinement. Finally, the communicative methodology can apply quantitative, mixed-methods and qualitative techniques [83, 86–89], and in the research presented here, a qualitative study with a communicative orientation was conducted.

### Study design, participants and context

**Previous context of the study.** The participants in the DGF were a group of acquaintances of whom some were friends or colleagues. Many of them are university researchers who had been taking part for many years in a regular dialogic intellectual gathering (DIG) discussing books that collect the theoretical and scientific bases of different disciplines (sociology,

education, neuroscience, and politics, among others). This dialogic intellectual gathering was usually held on site (face-to-face) with reduced participation online, but it became fully online during the lockdown caused by the COVID-19 pandemic. In this context, a small group of participants had the idea to organize a specific dialogic gathering of films (DGF) during confinement and proposed this to the rest of the group, and most of them decided to participate. Moreover, some of them invited other friends, relatives or people who they were confined with who had never participated in the prior dialogic intellectual gatherings. This situation facilitated increased diversity among participants. During the period analysed (between 14 March 2020 and 10 May 2020), an average of 60 people participated regularly in the DGF, with some days reaching up to 85 different participants.

**Running of the DGF and discussions that resulted from each film.** A dialogic gathering of films consists of organizing dialogic gatherings to discuss films (fiction and nonfiction) selected not for their cinematographic quality but for the relevance of the discussions that these films can trigger among participants in this particular moment. In the process of selecting the films, the contextual situation, and participants' needs–schedule compatibility with different professions and lifestyles during confinement, main interests–were considered. To facilitate participation, all the films discussed during this period were freely available on the Internet.

In the DGF developed during confinement, each participant watched the film whenever it was most suitable for him or her, and then, the participants met through a free digital platform solely for the discussion of the film on a previously agreed date and time, lasting one hour. At the beginning of each session, one of the participants, a different participant each time, briefly introduced the main issues featured in the film in a maximum of 5 minutes. It is relevant to highlight that the person presenting the film did not act as an expert on the film who decided what to discuss and what not to discuss; rather, the idea was to have someone introduce the film from a personal perspective to facilitate the start of the discussion, rather than having an expert explain important facts or features of each of the films. The person who introduced the film usually placed the film in a socio-historical moment and/or related it to a scientific, cultural and/or social event, highlighting some parts of the film he or she considered relevant while linking them with social, historical, cultural realities and/or relationships with other films, personal reflections and/or experiences or doubts. The topics for the issues to be discussed were not decided beforehand, but they emerged during the discussion through the dialogue, and one of the participants moderated the session.

Then, the moderator was responsible for giving the floor to all the participants who indicated a desire to enter the discussion that followed a turn to speak without interrupting any other participant. To participate in the discussion, participants asked for a turn by raising their hand on the online platform. The moderator ensured that everyone who wanted to participate could do so to ensure a diversity of participation in the discussion. The moderator also gave the floor first to those participants who had not yet taken the floor–before those who might want to contribute for a second time, to avoid monopolization of the debate by any person. People participating can refer to the issues stated by the person who presented the film or can raise new issues for discussion, without any obligation to follow any specific discussion. The diversity of profiles participating with different contributions promotes a rich and diverse discussion: the same film or scene can evoke different interpretations and can be related to very different social and personal realities.

This specific DGF was initiated on 14 March 2020 and was held every other day until 31 May 2020 for one hour each time. Later, when the confinement measures became more flexible, the DGF continued with a frequency of two days per week until the so-called *new normal* (describing the conditions after the lockdown ended) started in Spain on 21 June 2020. The

results discussed in this paper focus on the period between 14 March and 10 May. In Table 1, the dates of each DGF session and the name of the film discussed are introduced.

**Study design.** Considering the benefits that participants started to experience and discuss, a group of participants had the idea of developing a qualitative study [90] about the DGF experience to explore the benefits for participants.

The proposal of participating in this study was presented to all the participants in the DGF as an independent aspect of their participation in the DGF to ensure that free participation in the DGF was not associated with compulsory participation in the research proposal. In fact, the proposal was presented by an independent email sent to the participants and was not presented during a DGF session. Thus, this issue was never addressed over the course of any DGF session. It was only discussed with those participants who responded to the email and reported an interest in participating in the research. This process ensures free participation in the research and that there was no situation that the participants could have perceived as pressure to participate in the study, thereby ensuring compliance with ethical research standards [91].

**Table 1. Dialogic gathering of films.**

| DATE | FILM | DIRECTED BY |
| --- | --- | --- |
| 14/3/20 | Virus (2013) | Kim Sung-su |
| 15/3/20 | My Fair Lady (1964) | George Cukor |
| 17/3/20 | Claudio Monteverdi (1985) *Documentary Film* | Tony Cash |
| 19/3/20 | Citizen Kane (1941) | Orson Wells |
| 21/3/20 | An Enemy of the People (1981) *Televised theatre play* | Francisco Abad |
| 23/5/20 | Cotton Club (1984) | Francis Ford Coppola |
| 25/3/20 | To Live (1994) | Zhang Yimou |
| 27/3/20 | Baahubali: The Beginning (2015) | S.S Rajamouli |
| 29/3/20 | The Plague (1992) | Luis Puenzo |
| 31/3/20 | Romeo and Juliet (1968) | Franco Zeffirelli |
| 2/4/20 | Arabian Nights (2000) *Two part mini-series* | Steve Barron |
| 4/4/20 | La fille de Brest (2016) | Emmanuel Bercot |
| 6/4/20 | Romero (1989) | John Duigan |
| 8/4/20 | And the Violins Stopped Playing (1988) | Alexander Ramati |
| 10/4/20 | Florence Nightingale (2008) | Norman Stone |
| 12/4/20 | Ulises (1954) | Mario Camerini |
| 14/4/20 | Oliver Twist (1969) | Carol Reed |
| 16/4/20 | Ready Player One (2018) | Steven Spielberg |
| 18/4/20 | Artificial Intelligence. Our best friend (2018) *Documentary Film* | Blaise Piguet |
| 20/4/20 | Socrates (1970) | Roberto Rossellini |
| 22/4/20 | Lawrence of Arabia (1962) | David Lean |
| 24/4/20 | Why Beauty Matters. (2009) *Documentary Film* | Louise Lockwood |
| 26/4/20 | The Imitation Game (2014) | Morten Tyldum |
| 28/4/20 | Opera: Fidelio (Beethoven) | |
| 30/4/20 | Casablanca (1942) | Michael Curtiz |
| 2/5/20 | Doctor Zhivago (1965) | David Lean |
| 4/5/20 | Neuroscience Conference from Dr Leone *Conference* (2019) | Fundación Querer |
| 6/5/20 | Selma (2014) | Ava DuVernay |
| 8/5/20 | Watched (5G) (2019) *Documentary Film* | José Antonio Guardiola |
| 10/5/20 | The Nibelungs (1st part). The Death of Siegfried (1966) | Harald Reinl |

* Spanish titles have been translated into English.

Table 2. Gender and age distribution of participants.

| Age | 24–30 | 31–40 | 41–50 | 51–60 | 61–69 | Total |
|---|---|---|---|---|---|---|
| Female | 11 | 8 | 14 | 4 | 1 | 38 |
| Male | 0 | 6 | 6 | 0 | 3 | 15 |
| | | | | | n = | 53 |

The four authors of this paper are participants in the DGF and also acted as participants in the research and responded to the questionnaire.

**Participant profile.** As explained in greater detail in the section below, an online questionnaire was elaborated for data collection. Once revised and edited, the questionnaire was sent by email to a purposeful sample [92], namely, all the usual participants in the DGF, inquiring whether they wished to participate (voluntarily) and guaranteeing anonymity. Fifty-three people responded to the questionnaire and signed a consent form assuring their voluntary and anonymous participation. The 53 people were adults, 38 women and 15 men, between 24 and 69 years old. Table 2 presents the gender and age distribution.

Participants were of different nationalities; most of them were Spanish, and 4 of them identified themselves as Chilean, Romanian, Uruguayan and Brazilian. The majority of participants did so from Spain, while 3 participants joined from the UK, Brazil and the USA. The cities of residence and nationalities of the participants are presented in Table 3.

Regarding professional background, most participants (30) identified their profession within the scope of education: from primary education to university. The professional backgrounds reported by the participants are presented in Table 4.

## Data collection technique

For the data collection, an online questionnaire form was specifically created for this study. No existing questionnaires allow us to investigate the reality of DGFs; thus, a new questionnaire was created to study this new reality. To create a validated questionnaire, researchers elaborated the questionnaire following the dialogic design steps [83, 93] as follows: a) researchers asked participants about the main issues to highlight in the questionnaire; b) researchers elaborated an initial questionnaire proposal; and c) a group of participants in the DGF revised the questionnaire to ensure that it was understandable, collected information on the main issues, and ensured anonymity. They revised the questionnaire directly in their digital form to further ensure that the tool functioned correctly and to confirm that the entire questionnaire allowed free responses to any question. This external revision identified some mistakes in the online

Table 3. Country and place of residence of participants.

| Country | City of Residence | |
|---|---|---|
| Brazil | Sao Carlo | 1 |
| Spain | Basque Country (different towns) | 9 |
| | Catalonia (different towns) | 30 |
| | Santander | 2 |
| | Valencia (different towns) | 7 |
| | Castilla y León | 1 |
| | Zaragoza | 1 |
| United Kingdom | Cambridge | 1 |
| United States of America | Madison | 1 |
| | n = | 53 |

**Table 4. Professional backgrounds of participants.**

| Profession | Number of people |
|---|---|
| University | 17 |
| Teacher in primary education | 2 |
| Teacher in special education | 1 |
| Teacher in secondary education | 3 |
| Teacher in adult education | 2 |
| Teacher (not specified) | 3 |
| Education (not specified) | 2 |
| University student | 2 |
| Engineer | 2 |
| Programmer | 1 |
| Environmental sciences | 1 |
| Public administration | 1 |
| Social (not specified) | 1 |
| Retired | 2 |
| No response | 13 |
| | **n = 53** |

tool, for example, the obligation to respond to one question in the middle of the questionnaire before being able to proceed. This requirement was eliminated before the questionnaire was sent to all participants. Finally, d) researchers elaborated the final version of the questionnaire.

The questionary consists of 50 questions divided into 3 sections, with 31 open-ended questions and 19 closed-ended questions, 16 of which have a Likert scale ranging from 1 to 4. None of the questions was mandatory. The organization of the questions is presented in Table 5.

Regarding question content, Section 1 includes 5 demographic questions (gender, age, city of residence, nationality and professional field), in addition to 3 open questions and 2 closed-ended questions about the access to and reasons for participating in the DFG and the conditions for participation. Section 2 focused on the description of the DGF, including 10 open-ended questions and 11 closed-ended questions with a Likert scale ranging from 1 to 4 (with 1 being the lowest and 4 the highest). The questions related to functioning, film selection, diversity of participants, most significant films, the cultural, scientific and intellectual quality of discussions, freedom of expression, and respect for plurality. Section 3 focused on the impact of dialogic gatherings of films and includes 12 open questions and 7 closed-ended questions with a Likert scale from 1 to 4 (with 1 being the lowest and 4 the highest). The questions related to evaluating the films or debates that generated most reflections, identifying the films shared with others beyond the DGF and the films that were watched again, exploring the impact on the professional field and at the personal level and understanding the extent to which the DGF helped cope with the social situation generated by COVID-19.

**Table 5. Questionnaire organization.**

| | Open-ended questions | Closed-ended questions (two options or multiple choice) | Closed-ended questions (Likert scale) | Total |
|---|---|---|---|---|
| **Section 1. Dialogic Gathering of Films (Questions 1–10)** | 8 | 2 | 0 | **10** |
| **Section 2. Description of the Dialogic Gathering of Films (Questions 11–31)** | 11 | 0 | 10 | **21** |
| **Section 3. Impact of Dialogic Gathering of Films. (32–50)** | 12 | 1 | 6 | **19** |
| **Total** | **31** | **3** | **16** | **n = 50** |

## Data analysis

Once all the information was collected, the analysis was conducted through a *hand analysis* [94] to identify the benefits that the DGF had for the participants. The steps of the analysis were as follows: a) stablishing a large category of analysis, *benefits for participants*, divided into professional benefits, personal benefits and benefits related to the COVID-19 pandemic according to the communicative orientation related to social transformation; b) preliminary exploratory analysis, reading the information collected several times to obtain a general sense of the data [95, 96]. In this process, new categories emerged from the information collected following a "bottom-up" approach [94]; c) conducting pre-analysis while reading and identifying new categories by using colour coding and making comments in the margins related to the different benefits identified; d) copying into separate files the information divided into the different themes, for example grouping all the information related to "professional benefits"; e) pre-reporting findings, including participants' quotes and combining different forms of *narrative discussion* [94]: discussions describing events; discussions of themes and discussions about how participants were empowered or changed; and f) validating findings with participants by sending them the reported results [94]. The 53 participants received the results by email and had the opportunity to read and make any comments they considered necessary. According to the communicative methodology, participants are involved in the interpretation of the results [97]. Some of the participants advised the researchers of formal mistakes and/or parts that needed rewording, editing and/or improved explanation, and all the participants confirmed their agreement with the results highlighted. Finally, g) final reporting was done by the researchers. To validate the qualitative findings [94], the researchers used strategies such as *triangulation* to corroborate evidence from different researchers and *member checking* by the participants in the research not only by returning the results to the participants but also by following communicative methodology [75, 77, 78], establishing, as stated previously, a dialogue between participants and researchers throughout research process from the dialogic design of the questionnaire to final validation of the results.

## Ethics statement

The study was fully approved by the Ethics Board of the Community of Researchers on Excellence for All (CREA). The participants provided their informed written consent to participate in this study. The information provided in the consent form explained the goal of the study, the voluntary nature of participation, the ability to withdraw from it at any time, the data collection procedure, the materials and measures to be used, the permission to publish the data obtained, and the anonymity and privacy statement. Research participants had time to read the consent form and to ask the researchers questions, by email due to the pandemic situation.

## Results

In this section, we first present the main and/or more common discussions raised during the different DGF sessions and second the main results of the questionnaires focusing on the benefits that participation in this DGF had for the participants.

Some of the main and/or more common discussions raised during the different DGF sessions were social issues such as racism, inequality, feminism, poverty, gender violence, and human rights; specific discussions about the social situation linked to COVID-19; education in general and education linked to overcoming inequalities and improving learning; and historical, scientific and cultural advancements related to health, technologies, and social rights. In addition, personal reflections and shared experiences about family, social relationships and lifestyles in relation to health and technologies were present in the discussions. All of these

discussions that consistently respected highly diverse perspectives generated multiple benefits that are presented in the results section of this paper. In Table 6, some of discussion topics that resulted from each film are discussed.

The diversity of issues that were discussed generated various benefits for the participants. The impacts of the DGF related to its communicative orientation focused on social

**Table 6. Selected discussion topics that resulted from each film (general ideas).**

| FILM | DISCUSSIONS RESULTED (general ideas among others) |
|---|---|
| Virus (2013) | Human values, health, pandemics, science |
| My Fair Lady (1964) | Socialization, discrimination, learning, friendship, love, education |
| Claudio Monteverdi (1985) *Documentary Film* | Culture, love, knowledge, music |
| Citizen Kane (1941) | Human values, human rights, emotions and feelings |
| An Enemy of the People (1981) *Televised theatre play* | Discrimination, human values, social rights, politics, family |
| Cotton Club (1984) | Music, racism, violence, emotions and feelings (love, friendship) |
| To Live (1994) | Politics, culture, love, family, solidarity |
| Baahubali: The Beginning (2015) | Culture, music, human values, emotions and feelings, mythology, history |
| The Plague (1992) | Human values, human rights, friendship, health pandemics, literature, science |
| Romeo and Juliet (1968) | Love, friendship, emotions and feelings, literature, violence, feminism |
| Arabian Nights (2000) *Two part mini-series* | Culture, literature, feminism, gender violence, human values, history |
| La fille de Brest (2016) | Science, research, solidarity, human values, health |
| Romero (1989) | Religion, human values, human rights, social rights, history |
| And the Violins Stopped Playing (1988) | Racism, cultural minorities, war, Nazism, human rights, culture, music, discrimination, human values |
| Florence Nightingale (2008) | Health, feminism, solidarity, war, |
| Ulises (1954) | Mythology, literature, friendship, love, human values |
| Oliver Twist (1969) | Discrimination, poverty, child abuse, gender violence, solidarity |
| Ready Player One (2018) | Technology, love, friendship, solidarity, human rights |
| Artificial Intelligence. Our Best Friend (2018) *Documentary Film* | Technology, social relationships, health, education, science advances |
| Socrates (1970) | History, culture, social relationships, friendship, education |
| Lawrence of Arabia (1962) | History, friendship, discrimination, violence, interculturality |
| Why Beauty Matters. (2009) *Documentary Film* | Culture, Art, Beauty, history |
| The Imitation Game (2014) | Science advance, war, solidarity, history, sexual discrimination |
| Opera: Fidelio (Beethoven) | History, music, opera, feminism, social and human rights, inequalities |
| Casablanca (1942) | Politics, solidarity, friendship and love, history |
| Doctor Zhivago (1965) | Culture, politics, literature, human values and human rights, friendship, love, history |
| Neuroscience Conference from Dr Leone *Conference* (2019) | Education, science advance, neuroscience, health, human rights, politics |
| Selma (2014) | Human and social rights, racism, community, education, cultural minorities, discrimination |
| Watched (5G) (2019) *Documentary Film* | Technology, human rights, education, scientific advances |
| The Nibelungs (1st part). The Death of Siegfried (1966) | Culture, mythology, history, emotions and feelings, opera, music |

\* Spanish titles have been translated into English.

transformation are retrieved from Section 3 of the questionnaire. In this context, the main source of information came from the open questions, but some quantitative data were also reported to contribute to the description of benefits. We structured the presentation of these improvements as follows: 1) personal welfare and attitudes towards the management of confinement; 2) living together and online relationships; 3) motivation and creativity in the professional domain; and 4) openness to a diversity of perspectives and realities, which improves the understanding, argumentation and positioning in social, scientific and ethical debates.

## Personal welfare and attitudes concerning the management of confinement

The participants in the research explained that the DGF helped manage the confinement and generated a more optimistic attitude towards it, leading them to think in a more positive rather than a negative way. When evaluating how helpful participation in the DGF had been in managing the impacts of the COVID-19 pandemic, 86.5% of the participants in the questionnaire gave it the maximum score. The following is how one of the participants explained it:

> The confinement at home during these weeks has been relatively easy for me in a time of such hardship. (. . .) These weeks I have spoken to many people (. . .) and [realized that] everything was much more difficult for them because they could not make sense of what they were doing. Instead of focusing on it (the lockdown) as a positive action, understanding that they are doing what it needs to be done, (. . .) these other people I am telling you about have had a hard time; a nuisance, boredom, and they have mostly focused on complaining. A space like the DGF contributes very positively to focusing on the experience we're living from a more enriching perspective both for oneself and for those around us". (Male, 43)

Some of the participants were confined alone in their homes, so the DGF became a moment to meet and engage with others, and this meeting point benefitted their emotional welfare. As a participant explained, *[The DGF] has helped me to face the situation related to COVID-19 in a much better way. I live alone and meet people every two days, and discussing such interesting issues has been emotionally very gratifying.* (Male, 40) At the beginning of the confinement, many of the participants expected it to be very difficult, and then, with the DGF, they stated that they felt more cheerful:

> It has helped me a lot, because, at first, I thought that the confinement would be very long and hard. Knowing that you're going to participate every two days and with such quality interactions is really heartening. You always have that expectation of what the next film will be, what the introduction and the debate will be around today. . . (Female, 24)

The DGF gave meaning to many participants and provided them with a gratifying feeling in a very difficult situation. As one woman put it:

> I live alone, and I have virtually not left my home for two months, (. . .) and, actually, the fact of knowing that every two days there is a DGF makes me feel very good, it has been an interesting period in which I am learning a lot. When people ask me how I'm doing, I always say "very well", and the truth is that I owe a lot to these gatherings. (Female, 42)

Participants placed high value on these connections to discuss the films because they are not "just any" connections but imply in-depth discussions and reflections that give rise to personal welfare and in-depth personal reflections. When participants assessed the extent to

which participation in the DGF helped them in a personal domain, 94% of participants rated it with the maximum score. One of the participants highlighted how the debates about some of the films led her to ask herself how she wanted to live her life:

> The discussions have generated very deep reflections for me on a personal level, upon which one can eventually make decisions. For example, when I saw and discussed "Citizen Kane", I reflected deeply on how I would not want such a life, even if I had fame and money, and about that friendships need to be taken care of. Other debates have made me want to try harder to be a generous, supportive and caring person. (Female, 24)

## Living together and online relationships

Social connections are a very important issue during confinement, as noted previously. People living in the same home spend more time together than usual, and people living in different homes who are accustomed to meet regularly are now not allowed to do so. The DGF had an important influence on different kinds of relationships. In relation to people living together, participants explained that the DGF sessions became a special event in their homes and a routine in their daily lives that is considered in household planning. As one participant explained, they organize their daily routines with the DGF as a central event:

> (. . .) over the course of the gatherings, I forgot about the confinement. It was a special moment that I experienced together with my partner in an exciting way. I can say that the gatherings have become a temporal reference, marking routines to some extent: the day of the gathering we used to watch the film while commenting on it; before the gathering, we would get ready for it; after the gathering, we commented on the contributions made; and the day after, we planned when we were going to watch the next film. (Male, 38)

In fact, many participants expressed that every other day, they awaited the DGF sessions with great expectations, as if they were very special events. One of the participants stated, *"It helps me to be happier, lively and motivated. The gathering is like the special event of the day, so you are looking forward to it.* (Female, 26) The following quote shows the emotion that a participant felt awaiting the DGF: *My days are filled with meaning and expectation. Waiting for the meetings is like waiting for the weekend dance when I was a teenager: pure emotion!* (Female, 55)

It is also remarkable how participants shared the films discussed in DGF with other people beyond the DGF participants. Of the 34 films they discussed in the DGF, they shared 29 of them with other people–both within and beyond the household–who did not participate in the DGF, including family, friends and work colleagues. The reasons for sharing the films are diverse: to share knowledge and promote learning, to open up discussions with other people, hoping that the other people will also enjoy the films, and to share an experience that can be helpful to better manage the confinement situation, among others. In the different cases, through sharing the films, participants created more connections with people with whom they did not live.

In general, the DGF generated different and richer conversations at home and with people outside the home that are not merely focused on COVID-19 problems and/or on their daily life. As one participant explained:

> It has helped me to have richer conversations with my partner and my family and friends. To broaden my personal and cultural horizons. To introduce new elements of reflection

and research. To continue reflection on love, attraction, friendship, solidarity, prejudices, feminism, the future. (Female, 44)

Another participant considered that the conversations at her home were usually diverse, but with the DGF, the variety increased and generated new reflections that she would not have had without the DGF:

On a personal level, it has helped me to generate other types of conversations with my family, which I consider vital. It is true I think that in my house, the conversations tend to be very varied because of the professions we each develop and the interests we have; even so, I think that the DGF has taken us a step further and made us share spaces and reflections that perhaps we would never have had together. (Female, 27)

## Motivation and creativity in the professional domain

The professional domain is another of the main areas affected by confinement. Throughout the lockdown period, some people have continued working face-to-face in a difficult situation, and others have shifted to online work from their homes, some of them combining this with family care. In the professional field, the DGF also generated improvements for some of the participants. In fact, 76.5% of respondents gave the maximum score on the way in which the DGF has helped them in their professional field, and 81% of the respondents did so regarding how the DGF would help them in their professional field in the future. First, far from being less motivated in their jobs, participants commented that DGF had a positive influence on being more active and proactive in their jobs. As one of the participants noted, the DGF gives her *ideas and motivation for more personal and professional projects beyond this period*. (Female, 44)

Second, participants commented that the issues discussed in the debates on the films increased their knowledge and learning in a diversity of themes. The discussions related to different scientific, social and historical issues, and as one participant noted, the discussions increased collective and individual knowledge in different areas for all the participants: *It provides me with knowledge to contextualize social, scientific and human analyses. It also gives me intellectual security*. (Female, 44) The DGF also awakened new interests in new knowledge, and many participants stated new personal goals to read new books or to go in depth into the study of certain topics. As one participant explained, *for example, I would not have thought to read "The Song of the Nibelungs" before the debate on the film. Now I'm already thinking about when I will have time to read it*. (Male, 37)

One important feature to be stressed is the fact that all the participants highlighted that delving into such knowledge was not triggered by simply having viewed the films but by the discussions generated around the films and the advantages offered in this respect by the diversity of participants. As one participant stated, *what [the films] do is give rise to such a valuable contribution. In fact, some of us have already seen many films and documentaries, but the most interesting thing is all the knowledge gained in the debate*. (Female, 28) Many of the participants worked as teachers (in primary and secondary schools and in university) and were teaching online classes. The knowledge they gathered influenced the classes they are teaching with their students, transferring the knowledge acquired in the DGF to their classes and students:

It has given me tools to share knowledge and give greater depth to the discussions with the students. For example, the debate on "Selma" was very useful for me to address the issue of racism and segregation, and the debate on "The Plague" has helped me to analyse the current context and social reactions in comparison with the past. (Female, 28)

Beyond the knowledge acquired, the democratic organization of the DGF also benefited participants who have transferred it to their professional context. The DGF operates in a democratic way, promoting voluntary participation under equal conditions for all the participants. As one participant stated, it encouraged to transfer this democratic model of functioning to other contexts: *the model of free, respectful, and diverse participation about quality issues to improve the learning of all, and everyone is training me to better transfer it to my contexts*. As another participant also stated, *above all, it contributes to my ability to promote equality-based dialogues in other spaces and with other people* (Female, 60). Other participants created other DGFs, replicating the model in their professional contexts.

## Openness to a diversity of perspectives and realities improves understanding, argumentation and positioning in social, scientific and ethical debates

A key benefit identified by the participants in the DGF is that it contributed an openness to a diversity of perspectives and realities that improved comprehension, argumentation and positioning in social, scientific and ethical discussions.

The participants in the DGF highlighted that having shared a broad diversity of perspectives in the films' discussions promoted greater openness towards the plurality encountered in society and greater awareness of its relevance. As one participant stated:

It has helped me to be more aware of the existing diversity and its importance, the plurality in society and the need to preserve it. [and it has helped me] in knowing more about and better understanding the society we live in and the history that has marked our societies. (Female, 46)

This diversity of perspectives comes, on the one hand, from the diversity that the films address and the different arguments displayed in them. On the other hand, and more important, the openness to other realities comes from the diversity of the very people participating in the discussion, which is given the maximum score by 86.5% of participants. This diversity enabled interventions in the discussions to be done from different cultural and social backgrounds, religious beliefs and lifestyles, enriching the dialogues. Some participants explained that they listened to the interventions of other participants who made reflections and interpretations about the film in a way that they would have never made themselves. This diversity of perspectives is seen as opening one's mind to other possibilities, whether one agrees with them or not, but which in any case increases awareness of others' interpretations and points of view and promotes respect towards them:

To have a more open, more analytical, more respectful, more curious, more committed, more active insight with more ambition and more self-improvement insight. To have knowledge that helps to read the world better, and therefore, allows to contribute to improve it. (Female, 43)

The knowledge learned and the fact of opening up to different perspectives contributed to better understanding social, historic, scientific and ethical discussions. This understanding is not reduced to the situations that appear in the films but also to understanding past, present and future situations. This comprehension and openness to other perspectives is not only "new knowledge" but also contributes to taking a stance in ethical discussions with more arguments. As a participant stated: *Furthermore, the debates that have arisen helped me a lot to be*

*better informed on many ethical issues and to take a stance in many situations in a more secure and argued way.* (Male, 37)

Various participants also highlighted that the debates held in the gatherings provided them with enriched arguments and an improved ability to identify fake news in relation to COVID-19 and better arguments to counter it. As another participant noted, *[the gatherings] have helped me to develop a greater critical capacity in the face of the news and fake news that have been spreading, also allowing me to transfer that knowledge to my close environment.* (Female, 60)

## Discussion

Earlier research has highlighted that diversity among the profiles of people (gender, age, socio-economic situation) provides a different response to the pandemic and different levels of risk of being more negatively psychologically affected by confinement. Women, students and elderly people, among others [24, 44, 45], are some of the social groups that are more vulnerable to this. We found that women, identified in the abovementioned research as having a greater risk of being psychologically affected, are the ones who participated the most in the DGF. Thirty-eight women–from 24 to 69 years old–participated in the DGF compared to 15 men. The results presented above show that all of them benefited from participation, giving them more intellectual security, opening their minds to different ways of understanding the same reality, improving their relationships and acting against loneliness.

Research by Quian & Yara [46], analyses how a diversity of personality, ideology, and other such factors, can influence people's mentality during COVID-19, stating that opinions, concerns, and behaviour regarding COVID-19 varied significantly across different profiles, such as gender, age, education, and place of residence. People with many different profiles participated in the DGF. This diversity produced a variety of ideas and interpretations in relation to COVID-19 and in relation to the different themes generated during the discussion of the films. In fact, as stated above, when participants rated the diversity in the debates, 86.5% of participants gave the maximum score. In this sense, our research also indicates that diverse profiles have diverse opinions, concerns and behaviours. Moreover, the DGF confirmed that this diversity of personality, ideology, and personal and social background benefitted the DGF and all the participants. Such dialogue among a diversity of voices promotes openness to other realities and a fuller comprehension of the situation, and as participants explained, this diversity "opened their minds". In fact, this openness and deeper and broader understanding of the social, historical and scientific facts has also been identified in previous research about other dialogic gatherings [60–74]. In the specific case of this DGF, the findings highly that the gatherings provided participants with enriched arguments in relation to COVID-19. People receive a considerable amount of information about the pandemic, including fake news [98]. As the participants stated, the DGF also influenced their ability to identify fake news in relation to COVID-19.

Various authors have highlighted the relevance of creativity and innovation in developing actions that help to relieve the burdens that come with this pandemic [51]. The DGF was demonstrably a creative and innovative adaptation of previous very successful dialogic gatherings. Participants were able to translate the successful characteristics of dialogic gatherings in general and dialogic gatherings of films in particular into an online version capable of addressing the different needs of people, as schedules, themes that are interesting in the present context, and the diversity of ways of living in a common situation of confinement, among other aspects, to create a DGF specifically tailored to confinement. In so doing, they contributed to combatting social isolation with an activity that became essential for many of the participants.

According to the literature review, a positive attitude in relation to COVID-19 [21] contributes to having lower levels of psychological distress, and maintaining individuals' social and cultural activity is a key element to preserve psycho-social health [47] during confinement. The present research on this specific DGF corroborates the abovementioned literature, as participation in this social and cultural activity promoted a more positive attitude among the participants towards the situation provoked by the pandemic.

## Conclusions, limitations and further research

In a context of preoccupation about the negative effects of confinement on the psycho-social health of people who are confined, the growing literature exploring the topic [21–29] and the WHO's call [30] to avoid social isolation, the results of this study show that this specific DGF combatted social isolation according to qualitative responses of participants and helped them to cope with the situation generated by the COVID-19 pandemic. A total of 86.8% of respondents gave the highest score when asked about this issue. The DGF also initiated social, personal and professional benefits for the people taking part. Regarding the extent to which participation in the DGF helped at the professional level, 76.5% of the respondents gave the maximum score, and approximately 81% did so when asked about the extent to which this participation could help them professionally. Moreover, approximately 94% of participants gibe the maximum score when they were asked about the extent to which participating in the DGF helped them on a personal level.

This is the first study about a dialogic gathering of films, and it is very specific to the context of a DGF developed during confinement due to COVID-19. The in-depth analysis of this reality is facilitated by this specificity, but it also produces the limitations of this study. The participants were highly diverse, but none of them belonged to vulnerable/disadvantaged social groups; most of them were Spanish and/or participated in the DGF from Spain, and most of them had a professional field linked to education. The study presents the benefits of a specific virtual DGF developed during confinement due to COVID-19, but there are no findings related to other virtual DGFs and/or to DGFs developed in person. Therefore, the abovementioned limitations open up possibilities for further research on other virtual or in-person DGFs, developed in other contexts and countries, with people working in professional fields not related to education. Other dialogic gatherings, mainly dialogic literary gatherings, have been studied in different higher and low socio-economic contexts, as well as with vulnerable groups such as migrant women and people in prisons. Following these previous studies, it would be interesting to conduct further research on the implementation of DGFs in different contexts and specifically with the participation of vulnerable groups, including cultural minorities or people with low socio-economic status. Expanding this research will enable us to analyse common and different influences on participants, particularly with respect to their gender, age and other individual characteristics. On the other hand, the possible benefits of DGFs could be further explored, particularly regarding diverse topics. According to the large amount of knowledge shared during the DGF, benefits related to increased knowledge about historical, social and scientific facts could also be studied. Moreover, as the DGF promoted social interactions between people with diverse backgrounds, further research could also explore the potential benefits related to gender discrimination, social cohesion, critical thinking, and cultural and religious diversity. The films selected for the DGF were not chosen for their cinematographic quality but for the debates they were likely to provoke among this particular group of people at the particular time of confinement. Following these criteria, further research could also delve into the specific discussions that one or some of these films generated and/or more specifically about the discussions are that are triggered by a specific film in a

particular socio-historical moment. In a context where the hardships of the COVID-19 pandemic persist, the ways in which a DGF initiative can be spread to different social groups and contexts to help the population combat isolation and promote positive psycho-social effects is of the utmost importance.

Finally, we note that the social impact of the DGF in terms of benefits for the participants is highly recognized by the participants themselves in their responses and beyond. When the *new normality* started and continuing the DGF with the same regularity was not compatible with the daily lives of most of the participants, they were reluctant to end the DGF and sought alternatives. Therefore, the DGF organized for lockdown was concluded as such, and a "summer DGF" meeting began to be held weekly during the summer of 2020. After summer, restrictions persisted in various ways depending on the professions of participants and the contexts where they lived, but the DGF will continue to be held, albeit less frequently.

## Supporting information

**S1 Questionnaire. Tertulia dialógica de películas (Spanish).**
(PDF)

**S2 Questionnaire. Dialogic gathering of films (English).**
(PDF)

**S1 Dataset.**
(PDF)

## Author Contributions

**Conceptualization:** Maria Padrós-Cuxart, Roseli Rodrigues de Mello.

**Formal analysis:** Mimar Ramis-Salas, Elena Duque.

**Methodology:** Maria Padrós-Cuxart, Elena Duque.

**Visualization:** Maria Padrós-Cuxart.

**Writing – original draft:** Elena Duque.

**Writing – review & editing:** Mimar Ramis-Salas.

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
