## [Decision Letter · Decision Letter 0]

14 Apr 2021

PONE-D-21-01969

DIALOGIC GATHERING OF FILMS. Promoting meaningful online interactions during COVID19 confinement.

PLOS ONE

Dear Dr. Duque,

Thank you for submitting your manuscript to PLOS ONE. After careful consideration, we feel that it has merit but does not fully meet PLOS ONE’s publication criteria as it currently stands. Therefore, we invite you to submit a revised version of the manuscript that addresses the points raised during the review process.

In the revised version of the paper please consider a text proofreading. Please take a close look to the tables listed in the paper - please replace Table X in row 182 with its associated number. Please provide more information related the Communicative Methodology you have used in the paper and provide some arguments for this choice. In the questionnaire provided as supplementary materials there are a series of descriptive statistics regarding the respondents - please add the summary of this information in the paper, in addition to the information already provided in Table 2. Please add a validation for the questions for which the answers have been gathered using a Likert scale. Please consider the comments of the reviewers listed at the bottom of this email.

We look forward to receiving your revised manuscript.

Kind regards,

Camelia Delcea

Academic Editor

PLOS ONE

Journal Requirements:

4. Please update the title page within your main document. Please ensure that you include a list all authors and all affiliations as per our author instructions and clearly indicate the corresponding author.

5. Please include captions for your Supporting Information files at the end of your manuscript, and update any in-text citations to match accordingly. Please see our Supporting Information guidelines for more information: http://journals.plos.org/plosone/s/supporting-information

Reviewers' comments:

Reviewer's Responses to Questions

**Comments to the Author**

1. Is the manuscript technically sound, and do the data support the conclusions?

Reviewer #1: Partly

Reviewer #2: Yes

2. Has the statistical analysis been performed appropriately and rigorously? 

Reviewer #1: No

Reviewer #2: N/A

3. Have the authors made all data underlying the findings in their manuscript fully available?

Reviewer #1: No

Reviewer #2: Yes

4. Is the manuscript presented in an intelligible fashion and written in standard English?

Reviewer #1: Yes

Reviewer #2: Yes

5. Review Comments to the Author

Reviewer #1: I like the introduction of the topic. You provide a nice detailed explanation for your concept. Your method is clear and easy to follow. Your results section does not include an explanation of how the data was analyzed and there is no report of your inter-rater reliability. This is an essential part for qualitative research.

I like the conclusions made and examples provided to support them. However, you are missing an important piece to show that there is consistency between those evaluating the survey answers and the conclusions made. I cannot recommend this article until this information is included.

Reviewer #2: This research covers a relevant topic of research given the ongoing COVID-19 pandemic over more than the past one year. It contributes to the literature to understand coping mechanisms to address loneliness and isolation resulting from COVID.

Overall, there are many grammatical errors across the manuscript that need to be address. This affects readability and meaning in places. Suggest using active voice over passive where possible and write the results in past tense. The paper moves between past and present tense. These need to be thoroughly revised.

Regarding the structure of the manuscript: Parts of introduction can move to the methods section and parts of methods can be moved to the results or removed where repeated, specific comments provided below. Suggest having a separate section for conclusion and implications, at present it sits under Discussion. Please highlight any possible Limitations of the study.

Introduction

Line 62, p. 3. Correct the spelling of ‘South Korea’.

Line 80, p. 4. under ‘Dialogue of Gathering films’, please elaborate on the ‘vulnerable groups’ mentioned.

Line 99, p. 5. Please specify the ‘participant needs’ or move this part to the methods section.

Under section ‘Dialogue of Gathering films’, a lot of it can move to the methods section.

Methods

Suggest reducing the length of the methods.

Section 2.1. ‘Running of the DGF and discussion resulted from each film’ please change to past tense for uniformity.

Last paragraph on p.9 and first paragraph is more relevant to the results section, suggest moving it to the results section.

Line 195-197, p. 12. Suggest breaking the sentence or rephrase for readability.

Line 209, p. 12. Suggest rephrasing for clarity to, ‘Majority of participants joined from Spain while 4 participants participated from UK, Brazil and the USA.’

Line 211, p.12, suggest replacing the word ‘fields’ to ‘backgrounds’.

Table 3. can include demographic profile of participants such as sex of participants (Male/Female), background and other details, if available, as these are discussed in the Discussion.

Please provide details of facilitators – who were they, how were they identified.

Last paragraph p. 16, can be moved to the results or removed as it is already mentioned in the results section.

Results

Please provide some detail of the participant whose data are shared verbatim as quotes, this could include sex and age of participant in bracket next to the data quote.

Change the percentages provided to whole rather than in points, e.g. 94.2% can be 94% and 94.6% can be 95%.

The last data quote on p.24 is in italics, while the one above is not. Kindly revise for uniformity.

Discussion

The Discussion section at this stage requires more detail and needs to be contextualised within the wider literature. Suggest elaborating discussion section on the positive impact of DGF on loneliness and helping participants professionally.

6. PLOS authors have the option to publish the peer review history of their article (what does this mean?). If published, this will include your full peer review and any attached files.

Reviewer #1: **Yes: **Jill A. Yamashita

Reviewer #2: No

---

## [Author Response · Author response to Decision Letter 0]

8 Jun 2021

We thank the reviewers and the editor for the suggestions made and the opportunity to improve the manuscript. We have also edited the manuscript with AJE services.

In what follows we provide detailed information in which we have addressed the reviewer’s and editor’s comments citing the line number (when necessary). The changes are highlighted in yellow in the file labelled “Revised Manuscript with Track Changes”.

EDITOR RESPONSES

Following the editor indications we have revised the style requirements, we have edited the manuscript with AJE services, we have incorporated Supporting Information (questionnaires and data set) and we have included the Tittle page within the manuscript as required. In the following table we address other responses to specific comments from editor.

EDITOR: In the revised version of the paper please consider a text proofreading. RESPONSE: We have edited the document with AJE. The certificate it is attached.

EDITOR: Please take a close look to the tables listed in the paper - please replace Table X in row 182 with its associated number. RESPONSE: Thank you for this observation. All the tables have been revised and renumbered when necessary. 

EDITOR: Please provide more information related the Communicative Methodology you have used in the paper and provide some arguments for this choice. RESPONSE: Arguments for the choice of Communicative Methodology have been included in lines 105-122.

EDITOR: In the questionnaire provided as supplementary materials there are a series of descriptive statistics regarding the respondents - please add the summary of this information in the paper, in addition to the information already provided in Table 2. RESPONSE: According with this suggestion, descriptive statistics regarding the respondents (gender, age, country, city of residence and professional field) are presented in Tables 2, 3 and 4.

EDITOR: Please add a validation for the questions for which the answers have been gathered using a Likert scale. RESPONSE: Thank you for this comment that allows us to explain more about the methodology process. As we are analyzing a new reality not studied before, it was not possible to find a validated existent questionnaire useful for study the DGF. Therefore, researchers created a new questionnaire. The questionnaire was elaborated and validated following the dialogic design steps from the communicative methodology. This is explained in lines 217- 232. 

Reviewers' Responses:

Reviewer #1: 

R1: I like the introduction of the topic. You provide a nice detailed explanation for your concept. Your method is clear and easy to follow. Your results section does not include an explanation of how the data was analyzed and there is no report of your inter-rater reliability. This is an essential part for qualitative research. RESPONSE: The process of qualitative data analysis is now explained in section Data Analysis (lines 249-274), including the validation of the qualitative findings.

R1: I like the conclusions made and examples provided to support them. However, you are missing an important piece to show that there is consistency between those evaluating the survey answers and the conclusions made. I cannot recommend this article until this information is included. RESPONSE: According to the different comments of reviewers we have separated Conclusions into two sections “3. Discussion” and “4. Conclusions, limitations and further research”. In both sections we argue about the links among the literature review, the results and the conclusions made.

Reviewer #2: 

R2: This research covers a relevant topic of research given the ongoing COVID-19 pandemic over more than the past one year. It contributes to the literature to understand coping mechanisms to address loneliness and isolation resulting from COVID. Overall, there are many grammatical errors across the manuscript that need to be address. This affects readability and meaning in places. Suggest using active voice over passive where possible and write the results in past tense. The paper moves between past and present tense. These need to be thoroughly revised. RESPONSE: We have edited the document with AJE. The certificate it is attached.

R2: Regarding the structure of the manuscript: Parts of introduction can move to the methods section and parts of methods can be moved to the results or removed where repeated, specific comments provided below.Following these indications, we have moved some parts of the introduction to the methods section and parts of methods to the results section. RESPONSE: Thank you for this useful comment, with this changes the article is clearer.

R2: Suggest having a separate section for conclusion and implications, at present it sits under Discussion. Please highlight any possible Limitations of the study. RESPONSE: According to the different comments of the reviewers, we have separated the Conclusions into two sections “3. Discussion” and “4. Conclusions, limitations and further research”. In section 4 we have highlighted the limitations of the study linking it to the possibilities for further research.

Introduction

R2: Line 62, p. 3. Correct the spelling of ‘South Korea’. RESPONSE: This has been corrected, thank you. 

R2: Line 80, p. 4. under ‘Dialogue of Gathering films’, please elaborate on the ‘vulnerable groups’ mentioned. RESPONSE: We have further elaborated around the concept of “vulnerable groups” in lines 86-87.

R2: Line 99, p. 5. Please specify the ‘participant needs’ or move this part to the methods section. RESPONSE: Participants’ needs clarification has been included in lines 142-143 and all the part has been moved to the methods section.

R2: Under section ‘Dialogue of Gathering films’, a lot of it can move to the methods section. RESPONSE: We have moved this information to the methods section.

Methods

R2: Suggest reducing the length of the methods. RESPONSE: We have moved information from methods section to results section reducing the length of methods. At the same time, and following other revisions received, we have also moved parts from the introduction to the methods section and included more explanation in relation to methods section, as was requested in the review process. Therefore, the methods section has not been substantially reduced.

R2: Section 2.1. ‘Running of the DGF and discussion resulted from each film’ please change to past tense for uniformity. RESPONSE: We have edited this section.

R2: Last paragraph on p.9 and first paragraph is more relevant to the results section, suggest moving it to the results section. RESPONSE: Thank you for your suggestion, we have moved this paragraph to the results section.

R2: Line 195-197, p. 12. Suggest breaking the sentence or rephrase for readability. RESPONSE: We have edited the document rephrasing these lines. 

R2: Line 209, p. 12. Suggest rephrasing for clarity to, ‘Majority of participants joined from Spain while 4 participants participated from UK, Brazil and the USA.’ RESPONSE: We have rephrased these lines as suggested. Thank you

R2: Line 211, p.12, suggest replacing the word ‘fields’ to ‘backgrounds’. RESPONSE: We have replaced this word as you suggested. 

R2: Table 3. can include demographic profile of participants such as sex of participants (Male/Female), background and other details, if available, as these are discussed in the Discussion. Please provide details of facilitators – who were they, how were they identified. RESPONSE: Demographic profile of participants (male/female; age; country and place of residence, nationality and professional field) are now included in Tables 3, 4 and 5 and in lines 198-204; 207-210 and 212-214.

R2: Last paragraph p. 16, can be moved to the results or removed as it is already mentioned in the results section. RESPONSE: This paragraph has been removed and some information has been incorporated at the beginning of the results section.

Results

R2: Please provide some detail of the participant whose data are shared verbatim as quotes, this could include sex and age of participant in bracket next to the data quote. RESPONSE: We have included sex and age in brackets at the end of each verbatim quote. Thank you for this suggestion that allows to present the results more accurately. 

R2: Change the percentages provided to whole rather than in points, e.g. 94.2% can be 94% and 94.6% can be 95%. RESPONSE: We have changed the percentages according to this comment.

R2: The last data quote on p.24 is in italics, while the one above is not. Kindly revise for uniformity. RESPONSE: We have removed italics and revised uniformity of quotes.

Discussion

R2: The Discussion section at this stage requires more detail and needs to be contextualised within the wider literature. Suggest elaborating discussion section on the positive impact of DGF on loneliness and helping participants professionally. RESPONSE: According to the different comments of the reviewers, we have separated Conclusions in two sections: “Discussion” and “Conclusions, limitations and further research”. Positive impact on loneliness and helping participants professionally have been included.

---

## [Decision Letter · Decision Letter 1]

21 Jun 2021

DIALOGIC GATHERING OF FILMS. Promoting meaningful online interactions during COVID19 confinement.

PONE-D-21-01969R1

Dear Dr. Duque,

We’re pleased to inform you that your manuscript has been judged scientifically suitable for publication and will be formally accepted for publication once it meets all outstanding technical requirements.

Kind regards,

Camelia Delcea

Academic Editor

PLOS ONE

Additional Editor Comments (optional):

Reviewers' comments:

Reviewer's Responses to Questions

**Comments to the Author**

1. If the authors have adequately addressed your comments raised in a previous round of review and you feel that this manuscript is now acceptable for publication, you may indicate that here to bypass the “Comments to the Author” section, enter your conflict of interest statement in the “Confidential to Editor” section, and submit your "Accept" recommendation.

Reviewer #1: All comments have been addressed

2. Is the manuscript technically sound, and do the data support the conclusions?

Reviewer #1: Yes

3. Has the statistical analysis been performed appropriately and rigorously? 

Reviewer #1: Yes

4. Have the authors made all data underlying the findings in their manuscript fully available?

Reviewer #1: No

5. Is the manuscript presented in an intelligible fashion and written in standard English?

Reviewer #1: Yes

6. Review Comments to the Author

Reviewer #1: This revision is wonderful. You addressed all of my concerns and did a great job explaining what you did for your analyses.

There was a part in the discussion that was slightly ambiguous. For example, in line 505 to 506, you state "the findings highly that the gatherings provided participants with enriched arguments in relation to COVID-19"--I am not sure what you mean by this.

Table 6 should be reformatted so that it is easier to read. For example, there should be a gap or space in between each film and discussion topics, so it aligns better.

Otherwise, I think the article is ready.

7. PLOS authors have the option to publish the peer review history of their article (what does this mean?). If published, this will include your full peer review and any attached files.

Reviewer #1: **Yes: **Jill Yamashita

---

## [Editor Report · Acceptance letter]

1 Jul 2021

PONE-D-21-01969R1 

Dialogic gathering of Films. Promoting meaningful online interactions during COVID-19 confinement. 

Dear Dr. Duque:

I'm pleased to inform you that your manuscript has been deemed suitable for publication in PLOS ONE. Congratulations! Your manuscript is now with our production department. 

Kind regards, 

on behalf of

Dr. Camelia Delcea 

Academic Editor

PLOS ONE